# Radiative cooling to deep sub-freezing temperatures through a 24-h day–night cycle

Zhen Chen[1,*], Linxiao Zhu[2,*], Aaswath Raman[1] & Shanhui Fan[1]

Radiative cooling technology utilizes the atmospheric transparency window (8–13 μm) to passively dissipate heat from Earth into outer space (3 K). This technology has attracted broad interests from both fundamental sciences and real world applications, ranging from passive building cooling, renewable energy harvesting and passive refrigeration in arid regions. However, the temperature reduction experimentally demonstrated, thus far, has been relatively modest. Here we theoretically show that ultra-large temperature reduction for as much as 60 °C from ambient is achievable by using a selective thermal emitter and by eliminating parasitic thermal load, and experimentally demonstrate a temperature reduction that far exceeds previous works. In a populous area at sea level, we have achieved an average temperature reduction of 37 °C from the ambient air temperature through a 24-h day–night cycle, with a maximal reduction of 42 °C that occurs when the experimental set-up enclosing the emitter is exposed to peak solar irradiance.

[1] Ginzton Laboratory, Department of Electrical Engineering, Stanford University, Stanford, California 94305, USA. [2] Department of Applied Physics, Stanford University, Stanford, California 94305, USA. * These authors contributed equally to this work. Correspondence and requests for materials should be addressed to S.F. (email: shanhui@stanford.edu).

From fundamental thermodynamics considerations, high-efficiency conversion from heat to work requires both a high-temperature heat source and a low-temperature heat sink. The vast majority of energy conversion processes at present use our ambient surrounding here on Earth as the heat sink. On the other hand, outer space, at a temperature of 3 K, provides a much colder heat sink. Moreover, Earth's atmosphere has a transparency window in the wavelength range from 8 to 13 μm that coincides with the peak of the blackbody spectrum of typical terrestrial temperatures ~ 300 K, enabling the process of radiative cooling, that is, radiative ejection of heat from Earth to outer space, and hence the direct radiative access to this colder heat sink. Exploitation of radiative cooling therefore has the potential to drastically improve a wide range of energy conversion and utilization processes on Earth.

The study of radiative cooling has a long history[1–21]. It has been well known since ancient times, that a black radiator facing a clear night sky can reach sub-ambient temperature[16]. More recently, daytime radiative cooling under direct sunlight was demonstrated[17], where one used a specially designed radiator that reflects most of the sunlight, but radiates efficiently in the atmospheric transparency window. However, the demonstrated performance thus far has been rather limited. For night time cooling[3–10], in typical populous areas the demonstrated temperature reduction from ambient air is on the order of 15–20 °C. A temperature reduction of up to 40 °C has been demonstrated only at high-altitude desert locations with extremely low humidity[2]. For daytime cooling[17], the demonstrated temperature reduction is ~ 5 °C. An important open question then is: what is the fundamental limit of temperature reduction that can be achieved in typical populous areas on Earth?

In this paper, we first theoretically show that ultra-large temperature reductions up to 60 °C below ambient can be achieved. The key to such ultra-large temperature reduction is to use highly selective thermal emitter matched to the atmospheric transparency window, and to minimize parasitic heat losses. Experimentally, we demonstrate an apparatus, which exhibits continuous passive cooling throughout both day and night. In a 24 h day–night cycle in winter, the cooler is maintained at a temperature that is at least 33 °C below ambient air temperature, with a maximal temperature reduction of 42 °C, which occurs when the apparatus enclosing the cooler is exposed to peak solar irradiance.

## Results

### Theoretical analysis

To illustrate the pathway towards achieving ultra-large temperature reduction, we first consider the ideal case, where the atmosphere is 100% transparent at a particular wavelength range outside the solar spectrum. In such a case, an emitter that has non-zero emissivity within this wavelength range, and zero emissivity outside, will reach the temperature of outer space of 3 K in the absence of parasitic heat loss, since in such a case the emitter is undergoing thermal exchange only with outer space.

For a more realistic case, we perform the theoretical analysis as illustrated in Fig. 1, where the transmittance of the atmosphere is taken to be typical of Stanford, California in winter (Supplementary Fig. 2). Here for simplicity, we analyse the performance of night time cooling. The performance of night time cooling provides the upper bound for the performance during daytime, an upper bound that can be reached by completely suppressing solar radiation on the emitter.

The steady-state temperature of a radiative emitter is determined by the energy balance among three key components (Fig. 1a; see Supplementary Note 4 for detailed analysis): the

emitted thermal radiation from the sample ($Q_{sample}$), the absorbed thermal radiation from the atmosphere ($Q_{atm}$) and the parasitic heat losses ($Q_{parasitic}$) characterized by a heat transfer coefficient $h$. We consider three different emitters: a black emitter, a near-ideal selective emitter (Fig. 1b) that has unit emissivity inside the atmospheric transparency window (8–13 μm) and zero emissivity outside, and the actual selective emitter used in this work. In Fig. 1c, we plot the net flux,

$$Q_{net} = Q_{sample} - Q_{atm} - Q_{parasitic}, \qquad (1)$$

as a function of the temperature of the sample, $T_{sample}$. The steady-state temperature of the sample is reached when the net flux ($Q_{net}$) reaches zero. Here we fix the ambient temperature ($T_{ambient}$) to be 20 °C, and use a typical atmospheric transmittance at Stanford in winter (Supplementary Fig. 2). For each emitter, we consider two parasitic heat transfer coefficients: $h = 8\,\mathrm{W\,m^{-2}\,K^{-1}}$ represents a typical experimental set-up without sophisticated thermal design, while $h = 0\,\mathrm{W\,m^{-2}\,K^{-1}}$ represents an ideal case with perfect thermal insulation (see Supplementary Note 7 and Supplementary Fig. 5 for more intermediate $h$ values).

Figure 1c underlines two key features. First, with a substantial parasitic heat loss ($h = 8\,\mathrm{W\,m^{-2}\,K^{-1}}$), the difference in performance between the black and the selective emitter is relatively small. Both the black emitter and the near-ideal selective emitter are restricted to a temperature reduction $|\Delta T| \sim 10\,°C$. Second, when the parasitic heat loss is completely eliminated ($h = 0\,\mathrm{W\,m^{-2}\,K^{-1}}$), there is a very large difference in terms of performance between the black and the selective emitter. Whereas the temperature reduction of the black emitter is limited to $|\Delta T| \sim 20\,°C$, the near-ideal selective emitter achieves a far higher temperature reduction $|\Delta T| \sim 60\,°C$. Thus, to approach the fundamental limit on radiative cooling, both selective emitter and ultra-low parasitic heat loss are essential. These considerations, together with the need to suppress solar irradiance during the daytime, motivate our design of the experimental apparatus and the selective emitter.

### Experimental design

The experimental apparatus consists of a selective emitter surrounded by a vacuum chamber that is shielded from direct sunlight (Fig. 2a). The key here is to ensure that the selective emitter is thermally decoupled from the ambient air and the sun, but coupled to outer space through the atmospheric transparency window. The apparatus therefore has the following features. First, the parasitic heat losses, including the air conduction and convection, and the radiation and conduction from the backside of the selective emitter, are minimized with the use of a vacuum chamber (evacuated to a pressure of $10^{-6}$ Torr), which encloses 10 concentric reflective radiation shields, and 4 long-hollow ceramic pegs in addition to the thermal emitter. These pegs provide mechanical support, while minimizing conduction loss to the emitter (for thermal design details, see Supplementary Note 1). Second, the vacuum chamber is equipped with a ZnSe window with double-side anti-reflection coating. Such a window has high transmittance in the wavelength range of the atmospheric transparency window, which ensures radiative access from the selective emitter to outer space. Third, the direct and indirect solar irradiance onto the emitter is minimized by a combination of a shade that is placed vertically at the side of the chamber, and a mirror-cone that surrounds the ZnSe window on the chamber. The shade and mirror-cone ensure that the selective emitter itself is exposed to only diffuse sunlight during the time, when the apparatus is exposed to direct sunlight. The cone also restricts the angular range of the apparatus to around the zenith direction where the sky is most transparent, and hence serves to

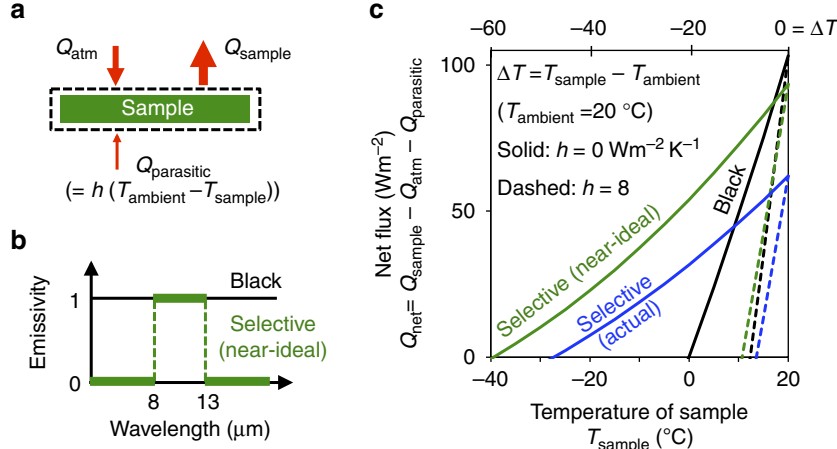

**Figure 1 | Theoretical analysis.** (**a**) Energy balance applied to the radiative emitter (dashed line). The net flux ($Q_{net}$) is determined by the outgoing flux from the emission of the sample ($Q_{sample}$), and the two incoming fluxes from the emission of the atmosphere ($Q_{atm}$) and the parasitic heat losses ($Q_{parasitic}$) characterized by a heat transfer coefficient $h$. (**b**) Three emitters are considered: a black emitter (black line), a near-ideal selective emitter (green line) and the actual selective emitter (blue line in Fig. 3b) of this work. (**c**) Net flux ($Q_{net}$) as a function of the temperature of the sample ($T_{sample}$). Note that the calculation is based on a typical atmospheric transmittance at Stanford in winter (grey line in Fig. 3b). The key parameter is the steady-state temperature corresponding to $Q_{net} = 0$. The analysis highlights the two key ingredients to achieve large temperature reduction below ambient: selectivity of the emitter and minimization of the parasitic heat loss. The performance of the actual selective emitter (blue) designed and tested in this work is also predicted.

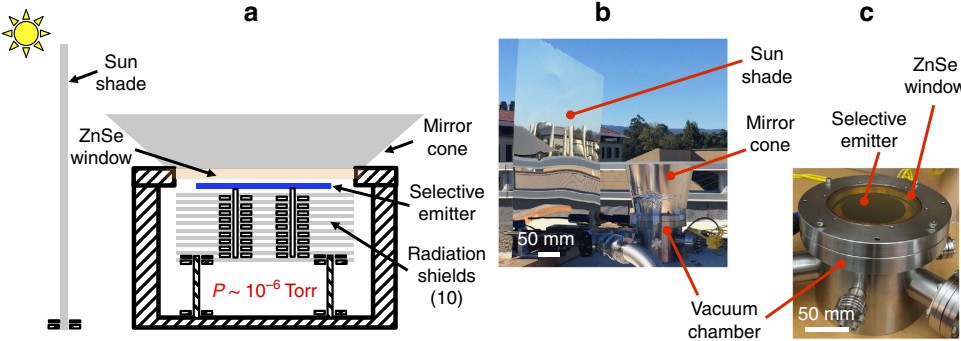

**Figure 2 | Experimental concept.** (**a**) Schematic of the experimental set-up. The key feature is to minimize parasitic heat losses of convection and air conduction using a vacuum system. Radiation shields and long-hollow ceramic pegs are exploited to further reduce the radiation and conduction losses through the backside of the selective emitter. The shinny sun-shade and mirror-cone are used to minimize solar irradiance. ZnSe is selected for its transparency in the mid-infrared wavelength range (red line in Fig. 3b). (**b**) *In situ* experimental set-up. (**c**) Details of the vacuum chamber, including the selective emitter and the ZnSe window.

prevent the relatively high-intensity incoming sky radiation from the low angles from reaching the selective emitter[22]. A photograph of the experimental apparatus placed on the roof is shown in Fig. 2b. Figure 2c shows a photograph of some details of the vacuum chamber, including the selective emitter and the ZnSe window.

The temperature of the selective emitter and the ambient air is measured by K-type thermocouples. Two thermocouples are anchored with conductive cement on the backside of the selective emitter: one at the centre and the other at the edge to check the temperature uniformity. The measured non-uniformity ($< 0.3$ K) is well within the resolution of the thermocouple.

Figure 3a shows a cross-sectional scanning electron microscope image of the emitter designed to approach the near-ideal emitter spectrum as shown in Fig. 1b. It consists of layers of silicon nitride ($Si_3N_4$), amorphous silicon (Si) and aluminium (Al), with thickness of 70, 700 and 150 nm, respectively (for details, see Supplementary Note 2), on top of a Si wafer underneath that provides mechanical support. Here, the emission arises primarily from the phonon–polariton excitation in $Si_3N_4$. Moreover, the

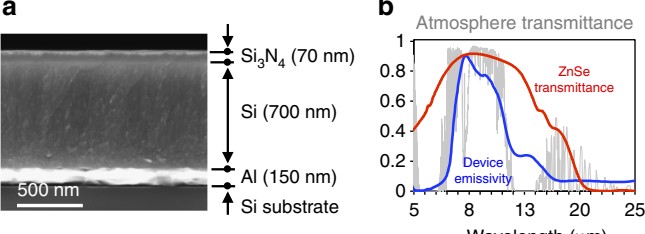

**Figure 3 | Structure and spectrum of the selective emitter.** (**a**) Cross-sectional scanning electron microscope image. (**b**) Spectral emissivity of the selective emitter (blue), measured using Fourier transform infrared spectroscopy and averaged over both polarizations, aligns very well with the atmospheric transmittance (grey). The ZnSe window (red) also confirmed to be transparent throughout the atmospheric transparency window. For clarity, here only show results along normal direction.

thickness of $Si_3N_4$ is chosen to be sufficiently small to substantially reduce unwanted radiative loss at wavelengths outside the atmospheric transparency window.

The emissivity spectrum of the structure is shown in Fig. 3b (blue line), together with the transmission spectrum of the ZnSe window (red; see Supplementary Note 3) of the vacuum chamber, and a typical local atmospheric transmittance (grey). Both the emissivity of the emitter and the transmission of the ZnSe window are characterized using Fourier transform infrared spectroscopy and averaged over both polarizations. We performed Fourier transform infrared spectroscopy characterizations on the selective emitter over the full hemispherical solid angles (Supplementary Fig. 1), but for clarity we only show data along the normal direction here. The emissivity exhibits a broad plateau that matches well with the atmospheric transparency window. Within this plateau the ZnSe window is also largely transparent. Therefore, the design here ensures that the selective emitter can exchange heat effectively with outer space through the ZnSe window and the atmosphere. In the meantime, the emitter has little emissivity outside the transparency window, which minimizes the heating effect of the downward radiation from the atmosphere.

**Experimental results**. We performed measurements by exposing the experimental apparatus to a clear sky throughout a 24-h day–night cycle at Stanford, California in winter (see Supplementary Note 9 and Supplementary Figs 7 and 8 for an experiment in summer). A typical measurement (Fig. 4a) shows the temperature of the selective emitter (blue), the ambient air (grey), as well as their difference (red). The solar irradiance (green; right axis) of a typical clear day in winter is also measured for reference. A few prominent features can be clearly recognized from Fig. 4a. First, the temperature of the selective emitter rapidly decreases to be 40 °C below ambient air within half hour after the vacuum chamber is pumped down to $10^{-5}$ Torr. Second, it tracks closely the trend of the temperature of the ambient air in the following 24 h, with an average temperature reduction from the ambient of 37.4 °C. Finally, the maximal temperature reduction from ambient (42.2 °C) appears, when the apparatus enclosing the cooler is exposed to peak solar irradiance. This seemingly counter-intuitive observation points to the effectiveness of the sun-shade/mirror-cone for blocking sunlight, and arises from the high contrast between the ambient air temperature and the dew point, when the solar irradiance reaches its peak.

Figure 4b,c compare the experimental results (blue points) of the selective emitter to the theoretical predictions (blue shaded area; for details, see Supplementary Notes 4 and 5 and Supplementary Fig. 3). A control experiment (black points) is also performed on a near-black emitter consisting of a 50 μm fused silica slide coated on its backside with 150 nm of aluminium film. The theoretical predictions for such a near-black emitter is shown as the grey shaded area. In the theoretical prediction, we bound the performance of either device under maximal and minimal parasitic heat loss and diffuse solar irradiance, estimated in Supplementary Notes 1 and 5. In Fig. 4b,c, we consider the temperature reduction as a function of dew point (ambient air temperature), while keeping the ambient air temperature (dew point) fixed. Note that the dew point is obtained from a weather station at Stanford (https://www.wunderground.com/us/ca/stanford). In general, the experiment agrees well with the theory. The few outliers in Fig. 4b may be related to invisible thin cloud coverage. We discuss the cloud effect in more details in Supplementary Note 8 and Supplementary Fig. 6. In both Fig. 4b,c, we see that the selective emitter is far better than the near-black emitter for the purpose of reaching a significant sub-ambient temperature under small thermal load. Also, for the same range of variation in parasitic heat loss, the variation in performance for the selective emitter is far greater compared with

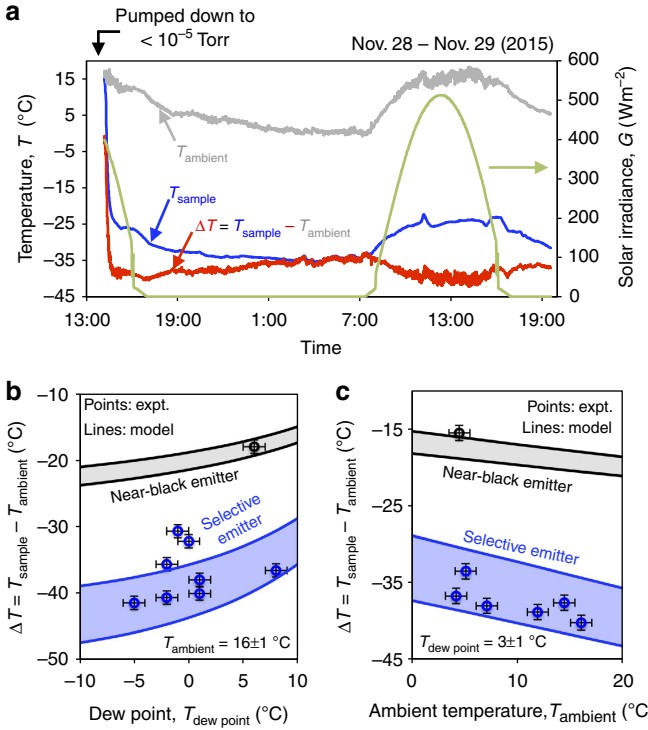

**Figure 4 | Experimental results.** (**a**) Large temperature reduction below ambient through radiative cooling in a 24-h day–night cycle. After pumping down to $10^{-5}$ Torr, the selective emitter rapidly cools down to ~40 °C below ambient temperature within half hour. A maximal cooling of 42.2 °C synchronizes with the peak of the solar irradiance, confirming the function of the sun-shade and the mirror-cone (Fig. 2), and also highlighting the high contrast between the ambient temperature and the dew point during this period. (**b**) Comparison with theoretical models: temperature reduction, $\Delta T$, as a function of dew point, with the ambient temperature fixed at $T_{ambient} = 16 \pm 1$ °C. (**c**) $\Delta T$ as a function of ambient temperature, with the dew point fixed at $T_{dew\text{-}point} = 3 \pm 1$ °C. Shaded areas, grey for near-black emitter and blue for selective emitter, represent the uncertainties of the model resulting from the uncertainties in estimating the parasitic heat losses and the absorption of the diffuse solar irradiance (for details, see Supplementary Notes 1 and 5). The error bar represents a conservative estimate on the sensitivity of our K-type thermocouples and the uncertainty in interpolating the dew point from a weather station at Stanford. Results in (**b**,**c**) underline a guideline to achieve large temperature reduction through radiative cooling: selective emitter with low dew point and high ambient temperature.

that of the near-black emitter. Thus, the selective emitter is more sensitive in its performance to the variation of parasitic heat loss, confirming the prediction shown in Fig. 1c.

Figure 4b shows that for a fixed ambient air temperature, the cooling performance improves as the dew point decreases. A low dew point results in a more transparent atmospheric window, and thus a better radiative cooling performance. Figure 4c shows that for a fixed dew point, the temperature reduction increases as the ambient temperature increases. Examining the energy balance of the emitter (equation 1), we see that the ambient temperature enters through both the downward atmospheric radiation ($Q_{atm}$) and the parasitic heat loss ($Q_{parasitic}$). On the other hand, the use of the selective emitter and the vacuum chamber significantly reduces these two terms, and as a result the dependence of the steady-state temperature of the sample on the ambient air temperature becomes weaker. Thus, although the lowest sample temperature occurs when the ambient temperature is low, the maximal temperature reduction occurs at the maximal ambient

temperature. To conclude from Fig. 4b,c, the maximal temperature reduction is obtained when the dew point is low and the ambient air temperature is high. In Fig. 4a, this occurs near the point of peak solar irradiance.

## Discussion

In summary, the experiments here provide a record-setting performance in radiative cooling during both day and night. The demonstrated steady-state temperature is far below the freezing point even though the apparatus enclosing the cooler is exposed to peak sunlight. Our work demonstrates the possibility of reaching the fundamental limit of radiative cooling by combining photonic and thermal design. From a practical point of view, radiative cooling is becoming important in a number of areas, including passive building cooling[23], renewable energy harvesting from the universe[24] and refrigeration in arid region[11]. Our current work points to an avenue for further improvement of radiative cooling systems, especially for applications where large temperature reduction is demanded, such as refrigeration in arid region[11]. For applications with large thermal load such as passive building cooling[23], however, a different selective emitter targeting at maximizing cooling flux should be pursued (see Supplementary Note 6 and Supplementary Fig. 4 for more details).

We end by briefly commenting on the prospect for scaling up the present system. Commercial evacuated solar water collectors[25] use the vacuum at a level that is comparable to what we use here. Therefore, the use of the vacuum itself is not an issue for scaling up. On the other hand, we do recognize several aspects of the current system that needs to be improved to achieve large-scale deployment of such high-performance radiative cooling systems. First of all, to achieve radiative cooling requires us to enclose the emitter with a material that is transparent to infrared wavelength range, while in the meantime is compatible with a vacuum system. The material we use, ZnSe, is too costly for this purpose. For large-scale deployment, one can envision the use of other materials such as Si or Ge, which are transparent in the wavelength range of 8–13 μm, as the window of the vacuum system. Importantly, these materials need not be transparent in the solar wavelength range. Second, it would be desirable to develop more robust shading scheme other than the sun-shade/mirror-cone that are currently used in our system. One strategy is to cover the vacuum system with infrared transparent solar absorbers to absorb the sunlight and dissipate the heat through natural convection. We envision future efforts along these directions to achieve a large industrial scale implementation of our current work.

**Data availability**. The data that support the findings of this study are available from the corresponding author on request.

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

## Acknowledgements

This work was supported by the Global Climate and Energy Project (GCEP) at the Stanford University. We thank Dr Eli Goldstein for supplying reflective metal sheets to make the sun-shade and the mirror-cone.

## Author contributions

S.F. conceived the idea and supervised the research. Z.C. conducted the thermal design and L.Z. conducted the photonic design. Z.C. and L.Z. performed the experiment, and interpreted the results. A.R. provided conceptual advice. Z.C. and S.F. wrote the paper. All authors commented on the manuscript.

## Additional information

**Publisher's note**: 

