## [Peer Review File · Nature Communications]

Reviewer #1 (Remarks to the Author)

The manuscript describes an experimental system to investigate the extreme temperature possibilities of radiative cooling devices. The system is novel in that it combines a skywindow selective radiating surface, a vacuum chamber system with ZnSe window, and a mirror cone / shading device to minimize incoming solar radiation. The experimental achievement of "ultra low" sub-ambient temperatures will be of great interest to many within the field and beyond.

I see no issues with the methodology, or experimental results, and I agree with the authors that this is an excellent result for the system, in achieving significantly lower sub-ambient temperatures than previous works, and that their work "points to an avenue for further improvement of radiative cooling systems". I do, however, note that these results are only achievable within very specific and narrow parameters (e.g very low thermal load and diffuse sunlight only through shading) and would not be achievable or scalable for purposes such as passive building cooling. This is not made sufficiently clear in the abstract, the discussion of results nor the author's conclusions. The paper merits publication, though I believe the following minor comments and revision suggestions ought to be considered and incorporated so as to avoid misleading readers.

Lines 22-23: I feel the abstract should end at "demonstrates significant fundamental potential for radiative cooling." As the rest of the sentence is likely to mislead readers as detailed later. This setup is designed for maximum temperature drop with minimum heat load, unlike building cooling which requires high heat-load capacity usually operating at a low delta T.

Line 56: should mention "indirect/diffuse peak solar radiation" to ensure that the reader does not misinterpret the result as having been achieved in full direct sunlight. Lines 53-54 could also read "...day and night in winter"; and Line 152 "A day-night cycle at Stanford, California in [add month]". Also lines 160-161; 199 "...even during peak sunlight"

Lines 76-78 / Fig 2: It would be good to see the inclusion of the performance data for the specific selective emitter that was tested, rather than just the theoretical data for a black body and an ideal selective emitter.

Figure 2: could be significantly improved by the inclusion of graph (possibly in the supplementary material) of the subset of Fig2c which covers just the region down to $\Delta T = -15$ {degree sign}C as this is the most likely operating region of radiative cooling devices with a cooling load. This would be of interest as it helps to demonstrate more clearly when a selective emitter would be preferred over a black body.

Lines 114-116: The author describes the mirror cone's purpose (along with the shade), as to minimize direct and indirect solar irradiation onto the emitter. The author appears to be unaware of the dominant purpose and more important benefit of the cone, which is that it allows the full hemisphere of outgoing thermal radiation, while limiting the relatively high intensity incoming sky radiation from the low angles. (Smith, G.B., "Amplified radiative cooling via optimised combinations of aperture geometry and spectral emittance profiles of surfaces and the atmosphere", *Solar Energy Materials and Solar Cells*, doi:10.1016/j.solmat.2009.05.015). A comment regarding the applicability or efficacy of this type of systems year round response should be added, to ensure that the results are not extrapolated inappropriately by readers of the manuscript. The radiative cooling system described is set up for use with low sun angles (such as in winter) - see figure 1a. It is relatively simple to set up a shading system that works well throughout the day to allow radiative cooling in shaded winter sun. It is a far greater challenge organising a shading system that can function during summer when the sun is high in the sky and cooling demand is at its greatest. Not only this, it also limits the availability of a clear view of the sky for radiative cooling. In practice this means active re-positioning of the shading structure could be required to ensure 24hour cooling in summer months.

Figure 4a : The figure shows only the global horizontal solar radiance. Whilst this is useful, the graph would be greatly improved if it also showed the diffuse solar component reaching the system. If the diffuse data is unavailable then an estimate of it should be made using TMY data for an equivalent day with a comment to such effect.

Lines 174-177 / Figure 4b: The "bump" in the cooling curve of Fig4b can most likely be attributed to an increase in thin cloud coverage. The author should comment on the effect of cloud coverage on the system to give readers a full understanding of how radiative cooling systems behave in non-ideal conditions. The influence of clouds on down-welling radiation is well known in the literature (for example: "Performance comparisons of sky window spectral selective and high emittance radiant cooling systems under varying atmospheric conditions.", Solar 2010 http://solar.org.au/papers/10papers/10_66_GENTLE.A.pdf) and "A method to measure total atmospheric long-wave down-welling radiation using a low cost infrared thermometer tilted to the vertical, Energy Volume 81, 1 March 2015, Pages 233-244) and can be easily monitored using a Pyrogeometer or infrared thermometer.

Lines 177-178 : The statement "we see that the performance of the selective emitter is far better as compared to the near-black emitter" is correct under the limited/following conditions intended to reach the maximum sub-ambient temperature; Though for clarity I would extend the statement to include something along the lines of: " ... in the case of a significantly sub-ambient temperature regime when there is a low heat-load." As at near sub-ambient or above ambient (e.g. cooling a building at night after the ambient temperature has dropped below the temperature of the building) a black body emitter can emit significantly more thermal radiation.

Lines 204-210: The use of such systems (incorporating shading devices and vacuum system with ZnSe windows) on a larger scale as mentioned in the concluding remarks is highly unlikely and seems overstated. I suggest the authors consider revising the conclusions which attempt to demonstrate the possibility of widespread usage of such systems by the use of broad sweeping vague claims:

The system demonstrates an excellent niche example of extreme temperatures that are possible with radiative cooling if the heat load is minimised, the extension of this to larger scale building cooling/power generation does not directly follow as implied in the concluding remarks (and the abstract). It is only in cases of very low parasitic heating (or thermal load) that significant subambient temperatures could be attained as is expected - passive refrigeration in arid regions is a more realistic/plausible suggestion. In the case of cooling a building the thermal load is such that the system will be operating in a regime much closer to ambient temperature so as to maximise the heat removal from the structure where, as demonstrated in Fig2 (and in previous literature) the use of a blackbody surface is preferred due to the higher radiative capacity.

When considering the W/m², particularly when talking along the lines of scalability, one must not only consider the footprint of the radiator but also the area required by ancillary equipment.

Lines 205-206: "The vacuum system can also be implemented on a large industrial scale". I would question this claim and the validity of the comparison to evacuated tube solar collectors (lines 206-208). This is a large stretch for a number of reasons: (1) this is a misleading comment as a non-expert in the field may not realise a significant difference, being that the evacuated tubes must be made from infrared transparent material (ZnSe) rather than Glass. Large areas of ZnSe evacuated tubes would make the system prohibitively expensive; (2) there would be potential environmental issues involved in having large amounts of ZnSe on rooftops. (3) Additionally, unlike evacuated tube solar collectors, the additional challenge of shading must be met (discussed earlier). While relatively simple for a small area, this becomes increasingly difficult as the areas are scaled up.

This is not to say this work doesn't offer "an avenue for improvement of radiative cooling

technology. However care ought to be exercised with wording around potential impacts and scalability.

Line 193: This line is likely to confuse readers. While the maximum delta T is achieved during daytime due to the increase in difference between effective sky temperature and ground level ambient temperature, it should clearly point out the absolute minimum temperatures are achieved during low-dew point and low ambient conditions.

Supplementary Material:

Line 313-315: Should explain how? "We further weaken the thermal contact between the ceramic pegs and the uppermost / lowermost radiation shields."

Line 347-352: Discusses the hemispherical emittance in the sky-window and outside the sky-window. The total hemispherical emittance should also be included.

Line 387: Discusses the absorbed power from the atmosphere, while neglecting the effect of the mirror cone and shading device which help to inhibit some of the incoming radiation.

In summary I look forward to seeing this manuscript published once these comments have been addressed as I feel the paper will be of great interest to a broad audience.

Reviewer #2 (Remarks to the Author)

In the present manuscript, Chen et al describe theory and experiments to radiatively cool an object significantly below the ambient and in fact significantly below (water's) freezing (point) during both night and day cycling. By combining thermal design and photonic design, they reach an average temperature reduction of 37 C compared to the surroundings when cycled over 24 hours and peak cooling of 42 C during daytime illumination. The main advantage over their previous work, Ref[14], is the use of better thermal management to reduce conduction and convection. They also present a new photonic structure with more selective emissivity. Overall, the work is well presented and shows a novel result: a factor of 8x better daytime radiative cooling. I do have a few comments and clarification that should be addressed, as described below.

- (1) "Ultra-high-Performance" in the title seems somewhat arbitrary. "High-performance" is adequate.
- (2) The authors say that they estimate the parasitic heat transfer coefficient, h , but more details are needed for how this was done. Without knowing how this is actually calculated the reader merely has to take the authors at their words.
- (3) Similar to the previous comment, in section A of the Supplementary Materials, h is quoted in the range of 0.1-0.3 Wm⁻²K⁻¹, but in section D it is quoted as 0.2-0.4 Wm⁻²K⁻¹.
- (4) It is mentioned that the selective emitter is composed of 700 nm of Si (see figure 3). Is the silicon crystalline or amorphous. I believe it is amorphous based on the deposition, but it should be stated as the dielectric properties could be different.
- (5) Figure 4b shows the change in temperature with Dew Point. Was the dew point measured or controlled? How?
- (6) There has been significant renewed interest in radiative cooling over the last few years, and the manuscript could benefit from mentioning some of the recent advances outside of building cooling. Examples include radiative cooling of solar cells [see: Zhu Optica 1, pp. 32-38 (2014) and Safi, Optics Express, pp. A1120-A1128 (2015)], reaching cryogenic temperatures in space [Youngquist, Optics Letters 41, pp. 1086-1089 (2016)], and radiative cooling of an air-cooled solar-thermal power cycle [Zeyghami, Energy Conversion and Management 106, pp. 10-20 (2015)].

Reviewer #3 (Remarks to the Author)

This study theoretically explores the practical limits of radiative cooling and experimentally demonstrates a radiative cooling system with much-improved performance. The paper is very well written and well put together. The results are clear, and the methodology sound and well outlined.

My main concern, though, is the novelty and the impact of the work, in particular, whether it warrants publication in Nature Communications. When it comes to radiative cooling, the results from the previous work from the same authors (ref 17) are both intriguing and very relevant. In [17] the same authors showed something that was not at all expected or even intuitive (at least to a non-specialist): namely, they showed that daylight radiative cooling is possible in a system with a relatively rudimentary thermal insulation from the environment (and certainly in a system without any evacuation of the surrounding air).

With this result established, the fact that an object should radiatively cool even more in vacuum (that is, when the heat conduction channel is further suppressed) is quite straightforward. Even though the emitter structure in [17] is somewhat different from the one in this work (for example, it seems to have higher emissivity above 15 μ m), I think the authors would agree that the structure from [17] would also substantially cool down if also placed in a vacuum environment.

It implies that, by and large, the main contributor to added cooling performance (and the main difference from previous works) is the introduction of vacuum to suppress parasitic heat losses. To me this is mostly a technical improvement, meaning that the paper seems short of reaching the novelty and the impact threshold needed for publication in Nature Communications. Additional means or reducing the heat loss have either been used for radiative cooling previously (e.g. acrylic/polystyrene to reduce heat conduction in [17] similar to ceramic pegs here) or are industry standards (e.g. radiation shields/shades/mirrors).

I would like to offer several comments and additions that may improve an otherwise very sound study. While these additions won't necessarily change the issues of novelty and impact, I hope the authors might find them useful.

- Figure 2c very nicely shows the difference between an ideal ($h=0$) and a non-ideal ($h=8$) parasitic losses case. It would be helpful to also show some intermediate cases of $0 < h < 8$, to allow for visualization of the progression of improved cooling with the reduction in heat loss. This could maybe be an additional supplementary figure.
- A comment on the relationship between the vacuum pressure and the value of h (or, maybe an estimate for their system) could be useful.
- The authors focus on the equilibrium temperature of the cooled device. However, an analysis of radiative cooling power (or extracted heat power) would be a nice addition to this work. If this is not immediately applicable to the geometry of their system, the authors may consider alternative cases where the cooling emitter is in direct contact (at least on one side) with the object intended to be cooled down.
- (minor) On line 207, a system of "solar water collectors" is referenced. A direct reference (instead of a reference of a reference, such as [23]) would improve "bookkeeping", but this is a minor point.

RESPONSE TO DECISION LETTER

Nature Communications Manuscript: NCOMMS-16-13526

Title: Ultrahigh-Performance Radiative Cooling Through a 24-hour Day-night Cycle

Authors: Z. Chen, L. Zhu, A. Raman, and S. Fan

In this Response Letter, we first give a summary of the major revisions, followed by a detailed point-by-point response to each of the reviewers' comments.

SUMMARY OF MAJOR REVISIONS

We have made extensive revisions, including

- Additional experimental results demonstrating radiative cooling in a 24-hour day-night cycle in summer.
- 4 pages, 6 figures, and 8 equations of additional clarification and theoretical analysis in the Supplementary Materials.
- 10 new references in the main text and the Supplementary Materials.

Specifically in response to Reviewer # 1, the key revisions include:

- (A) Additional experimental results in the Supplementary Materials, showing the cooling performance in summer, as well as the effect of cloud.
- (B) Additional theoretical analysis in the Supplementary Materials, clarifying the effect of the mirror-cone, as well as the effect of diffuse solar irradiance.
- (C) Clarification and discussion in the main text regarding the applicability and scalability of this work.

Specifically in response to Reviewer # 2, the revisions address all requests:

- (D) Additional details in the Supplementary Materials, clarifying how we estimate the parasitic heat transfer coefficient.
- (E) Additional up-to-date references to reflect in the main text to reflect the significant renewed interest in radiative cooling.

Specifically in response to Reviewer # 3, we emphasize that the key novelty of this work is to **theoretically predict the fundamental limit, and experimentally demonstrate a new record of radiative cooling**. The photonic design and thermal design are key requirements in order to achieve this goal. Other revisions include:

- (F) Additional details in the Supplementary Materials, clarifying the relationship between vacuum level and the parasitic heat loss.

(G) Additional figures in the Supplementary Materials, showing the gradual improvement of cooling performance with the gradual reduction in parasitic heat loss.

(H) Additional analysis in the Supplementary Materials, clarifying the current selective emitter is designed for reaching a fundamental low temperature, instead of applications with high cooling power. Thus the measurement of cooling flux is less interesting.

POINT-BY-POINT RESPONSE: On following pages.

Our responses are in blue italic font.

Revised manuscript text is underlined.

~~~~~  
~~~~~ **Reviewer: 1** ~~~~~  
~~~~~

The manuscript describes an experimental system to investigate the extreme temperature possibilities of radiative cooling devices. The system is novel in that it combines a skywindow selective radiating surface, a vacuum chamber system with ZnSe window, and a mirror cone / shading device to minimize incoming solar radiation. The experimental achievement of "ultra low" sub-ambient temperatures will be of great interest to many within the field and beyond.

I see no issues with the methodology, or experimental results, and I agree with the authors that this is an excellent result for the system, in achieving significantly lower sub-ambient temperatures than previous works, and that their work "points to an avenue for further improvement of radiative cooling systems". I do, however, note that these results are only achievable within very specific and narrow parameters (e.g very low thermal load and diffuse sunlight only through shading) and would not be achievable or scalable for purposes such as passive building cooling. This is not made sufficiently clear in the abstract, the discussion of results nor the author's conclusions. The paper merits publication, though I believe the following minor comments and revision suggestions ought to be considered and incorporated so as to avoid misleading readers.

**RESPONSE:**

*We thank the reviewer for the favorable high-level evaluation of the potential of this work, and for many insightful comments. Below we provide a point-by-point response to the reviewers' comments.*

**(R1 query 1)**        Lines 22-23: I feel the abstract should end at "demonstrates significant fundamental potential for radiative cooling." As the rest of the sentence is likely to mislead readers as detailed later. This setup is designed for maximum temperature drop with minimum heat load, unlike building cooling which requires high heat-load capacity usually operating at a low delta T.

**RESPONSE:**

*We agree. The revised abstract now ends the way the referee suggested.*

**(R1 query 2)**        Line 56: should mention "indirect/diffuse peak solar radiation" to ensure that the reader does not misinterpret the result as having been achieved in full direct sunlight. Lines 53-54 could also read "...day and night in winter"; and Line 152 "A day-night cycle at Stanford, California in [add month]". Also lines 160-161; 199 "...even during peak sunlight"

**RESPONSE:**

*In our setup, the radiative surface is indeed shielded from direct sun light. On the other hand, the overall system is exposed to direct sunlight. In response to the reviewer's comments, we have revised the main text, such that the distinction is explicitly made.*

**REVISED TEXT in Line 54-57:**

“In a 24 hour day-night cycle in winter, the cooler is maintained at a temperature that is at least 33 °C below ambient air temperature, with a maximal temperature reduction of 42°C, which occurs when the apparatus enclosing the cooler is exposed to peak solar irradiance.”

*REVISED TEXT in Line 159-161:*

“We performed measurements by exposing the experimental apparatus to a clear sky throughout a 24 hour day-night cycle at Stanford, California in winter (see Supplementary Materials I for an experiment in summer).”

*REVISED TEXT in Line 169-170:*

“Finally, the maximal temperature reduction from ambient (42.2 °C) when the apparatus enclosing the cooler is exposed to peak solar irradiance.”

*REVISED TEXT in Line 211-213:*

“The demonstrated steady-state temperature is far below the freezing point even though the apparatus enclosing the cooler is exposed to peak sunlight.”

*In addition, we have added a sentence in Line 119-121:*

“The shade and mirror cone ensure that the selective emitter itself is exposed to only diffuse sunlight during the time when the apparatus is exposed to direct sunlight.”

**(R1 query 3)** Lines 76-78 / Fig 2: It would be good to see the inclusion of the performance data for the specific selective emitter that was tested, rather than just the theoretical data for a black body and an ideal selective emitter.

*RESPONSE:*

*We have revised Fig. 2 to include the calculation for the selective emitter in this work.*

**(R1 query 4)** Figure 2: could be significantly improved by the inclusion of graph (possibly in the supplementary material) of the subset of Fig2c which covers just the region down to  $\Delta T = -15$  {degree sign}C as this is the most likely operating region of radiative cooling devices with a cooling load. This would be of interest as it helps to demonstrate more clearly when a selective emitter would be preferred over a black body.

*RESPONSE:*

*We have added a figure (Fig. S4) and a corresponding section (Supplementary Materials F) to discuss this application regime with a cooling load.*

**(R1 query 5)** Lines 114-116: The author describes the mirror cone's purpose (along with the shade), as to minimize direct and indirect solar irradiation onto the emitter. The author appears to be unaware of the dominant purpose and more important benefit of the cone, which is that it allows the full hemisphere of outgoing thermal radiation, while limiting the relatively high intensity incoming sky radiation from the low angles. (Smith, G.B., "Amplified radiative cooling via optimised combinations of aperture geometry and spectral emittance profiles of surfaces and the atmosphere", Solar Energy Materials and Solar Cells, doi:10.1016/j.solmat.2009.05.015).

*RESPONSE:*

*We thank the reviewer for this very insightful comment. We agree.*

*CHANGES:*

*We have added a sentence to this effect in the main text, with a citation to this excellent paper from Professor Smith. The added sentence reads:*

**“The cone also restricts the angular range of the apparatus to around the zenith direction where the sky is most transparent, and hence serves to prevent the relatively high intensity incoming sky radiation from the low angles from reaching the selective emitter22”.**

**(R1 query 6)** A comment regarding the applicability or efficacy of this type of systems year round response should be added, to ensure that the results are not extrapolated inappropriately by readers of the manuscript. The radiative cooling system described is set up for use with low sun angles (such as in winter) - see figure 1a. It is relatively simple to set up a shading system that works well throughout the day to allow radiative cooling in shaded winter sun. It is a far greater challenge organising a shading system that can function during summer when the sun is high in the sky and cooling demand is at its greatest. Not only this, it also limits the availability of a clear view of the sky for radiative cooling. In practice this means active re-positioning of the shading structure could be required to ensure 24hour cooling in summer months.

*RESPONSE:*

*We have conducted additional experiment in the summer time, where we have achieved a 24-hour day-night cooling in summer, with an average temperature reduction of 27 °C and a maximal of 37 °C. The shading structure remains fixed during the entire duration of the experiment.*

*CHANGES:*

*We have added two supplementary figures (Fig S7 and Fig. S8) and a corresponding section (Supplementary Materials I) to describe this summer experiment.*

*We have also added a sentence referring to the summer experiment in the main text, which reads:*

**“We performed measurements by exposing the experimental apparatus to a clear sky throughout a 24 hour day-night cycle at Stanford, California in winter (see Supplementary Materials I for an experiment in summer).”**

**(R1 query 7)** Figure 4a: The figure shows only the global horizontal solar radiance. Whilst this is useful, the graph would be greatly improved if it also showed the diffuse solar component reaching the system. If the diffuse data is unavailable then an estimate of it should be made using TMY data for an equivalent day with a comment to such effect.

*RESPONSE:*

*We thank this reviewer for pointing out this diffuse solar component to us. Combined with the effect of the mirror-cone in query 5 and 15, this diffuse solar component plays an important role in modeling the temperature reduction as shown in Fig. 4b-c of the main text.*

*We did not measure the diffuse data. We did not find the TMY data for Stanford, either. Instead, we followed a textbook model to deduce the diffuse component from our measured total solar irradiance in Fig. 4a. We further estimated the portion of this diffuse solar irradiance that is absorbed by the selective emitter.*

*CHANGES:*

*We have added one supplementary figure (Fig. S3) and a corresponding section (Supplementary Materials E) to update our heat transfer model to include this diffuse solar component.*

*Based on the effect of this diffuse solar irradiance, as well as the effect of the sun-shade and mirror-cone in query 5 & 15, we have updated our modeling prediction as shown in the shaded bands in Fig. 4b-c. Note that we have also corrected a prior bug used in our numerical code, which affects Fig. 4b-c.*

**(R1 query 8)** Lines 174-177 / Figure 4b: The "bump" in the cooling curve of Fig4b can most likely be attributed to an increase in thin cloud coverage. The author should comment on the effect of cloud coverage on the system to give readers a full understanding of how radiative cooling systems behave in non-ideal conditions. The influence of clouds on down-welling radiation is well known in the literature (for example: "Performance comparisons of sky window spectral selective and high emittance radiant cooling systems under varying atmospheric conditions.", Solar 2010 [http://solar.org.au/papers/10papers/10\\_66\\_GENTLE.A.pdf](http://solar.org.au/papers/10papers/10_66_GENTLE.A.pdf) and "A method to measure total atmospheric long-wave down-welling radiation using a low cost infrared thermometer tilted to the vertical, Energy Volume 81, 1 March 2015, Pages 233-244) and can be easily monitored using a Pyrogeometer or infrared thermometer.

*RESPONSE:*

*We have double-checked and confirmed that the weather of these outliers in Fig. 4b is clear. We didn't recall that we saw cloud when we conducted the measurement for these outliers, either. We suspect this might be related to some invisible cloud in very high altitude.*

*We have briefly commented the effect of cloud in the main text, and added a supplementary figure (Fig. S6) and a corresponding section (Supplementary Materials H) to discuss this effect in more details.*

*REVISED TEXT:*

**"In general, the experiment agrees well with the theory. The few outliers in Fig. 4b may be related to invisible thin cloud coverage. We discuss the cloud effect in more details in Supplementary Materials I."**

**(R1 query 9)** Lines 177-178 : The statement "we see that the performance of the selective emitter is far better as compared to the near-black emitter" is correct under the limited/following conditions intended to reach the maximum sub-ambient temperature; Though for clarity I would extend the statement to include something along the lines of: " ... in the case of a significantly sub-ambient temperature regime when there is a low heat-load." As at near sub-ambient or above ambient (e.g. cooling a building at night after the ambient temperature has dropped below the temperature of the building) a black body emitter can emit significantly more thermal radiation.

*RESPONSE:*

*We agree and has made the revision accordingly.*

*REVISED TEXT:*

**"In both Figs. 4b and 4c, we see that the selective emitter is far better than the near-black emitter for the purpose of reaching a significant sub-ambient temperature under small thermal load."**

**(R1 query 10)** Lines 204-210: The use of such systems (incorporating shading devices and vacuum system with ZnSe windows) on a larger scale as mentioned in the concluding remarks is highly unlikely and seems overstated. I suggest the authors consider revising the conclusions which attempt to demonstrate the possibility of widespread usage of such systems by the use of broad sweeping vague claims:

The system demonstrates an excellent niche example of extreme temperatures that are possible with radiative cooling if the heat load is minimised, the extension of this to larger scale building cooling/power generation does not directly follow as implied in the concluding remarks (and the abstract). It is only in cases of very low parasitic heating (or thermal load) that significant subambient temperatures could be attained as is expected - passive refrigeration in arid regions is a more realistic/plausible suggestion. In the case of cooling a building the thermal load is such that the system will be operating in a regime much closer to ambient temperature so as to maximise the heat removal from the structure where, as demonstrated in Fig2 (and in previous literature) the use of a blackbody surface is preferred due to the higher radiative capacity.

When considering the W/m2, particularly when talking along the lines of scalability, one must not only consider the footprint of the radiator but also the area required by ancillary equipment.

**(R1 query 11)** Lines 205-206: "The vacuum system can also be implemented on a large industrial scale". I would question this claim and the validity of the comparison to evacuated tube solar collectors (lines 206-208). This is a large stretch for a number of reasons: (1) this is a misleading comment as a non-expert in the field may not realise a significant difference, being that the evacuated tubes must be made from infrared transparent material (ZnSe) rather than Glass. Large areas of ZnSe evacuated tubes would make the system prohibitively expensive; (2) there would be potential environmental issues involved in having large amounts of ZnSe on rooftops. (3) Additionally, unlike

evacuated tube solar collectors, the additional challenge of shading must be met (discussed earlier). While relatively simple for a small area, this becomes increasingly difficult as the areas are scaled up.

This is not to say this work doesn't offer "an avenue for improvement of radiative cooling technology. However care ought to be exercised with wording around potential impacts and scalability.

*RESPONSE to QUERIES 10 and 11:*

*We agree with this reviewer regarding their concerns on the applicability and scalability of our current design.*

*CHANGES:*

*We have revised the last paragraph of the main text to explicitly comment on these issues.*

*REVISED TEXT in response to QUERIES 10 and 11:*

"... Our current work points to an avenue for further improvement of radiative cooling systems, especially for applications where large temperature reduction is demanded, such as refrigeration in arid region11. For applications with large thermal load such as passive building cooling24, however, a different selective emitter targeting at maximizing cooling flux should be pursued (See Supplementary Materials F for more details).

We end by briefly commenting on the prospect for scaling up the present system. Commercial evacuated solar water collectors26 use the vacuum at a level that is comparable to what we use here. Therefore, the use of the vacuum itself is not an issue for scaling up. On the other hand, we do recognize several aspects of the current system that needs to be improved in order to achieve large-scale deployment of such high performance radiative cooling systems. First of all, to achieve radiative cooling requires us to enclose the emitter with a material that is transparent to infra-red wavelength range, while in the meantime is compatible with a vacuum system. The material we use, ZnSe, is too costly for this purpose. For large-scale deployment, one can envision the use of other materials such as Si or Ge, which are transparent in the wavelength range of 8-13 micron, as the window of the vacuum system. Importantly, these materials need not be transparent in the solar wavelength range. Second, it would be desirable to develop more robust shading scheme other than the sun-shade / mirror-cone that are currently used in our system. One strategy is to cover the vacuum system with infrared transparent solar absorbers to absorb the sunlight and dissipate the heat through natural convection. We envision future efforts along these directions in order to achieve a large industrial scale implementation of our current work."

**(R1 query 12)** Line 193: This line is likely to confuse readers. While the maximum delta T is achieved during daytime due to the increase in difference between effective sky temperature and ground level ambient temperature, it should clearly point out the absolute minimum temperatures are achieved during low-dew point and low ambient conditions.

*RESPONSE:*

*We have revised the text to make this point more clear.*

*REVISED TEXT:*

“... and as a result the dependence of the steady state temperature of the sample on the ambient air temperature becomes weaker. Thus, although the lowest sample temperature occurs when the ambient temperature is low, the maximal temperature reduction occurs at the maximal ambient temperature. To conclude from Figs. 4b and 4c, the maximal temperature reduction is obtained when the dew point is low and the ambient air temperature is high.”

Supplementary Material:

**(R1 query 13)** Line 313-315: Should explain how? "We further weaken the thermal contact between the ceramic pegs and the uppermost / lowermost radiation shields."

*RESPONSE:*

*We revised this sentence to make it more clear.*

*REVISED TEXT in Supplementary Materials:*

**“We further weaken the thermal contact between the ceramic pegs and the uppermost / lowermost radiation shields by roughening the contact areas.”**

**(R1 query 14)** Line 347-352: Discusses the hemispherical emittance in the sky-window and outside the sky-window. The total hemispherical emittance should also be included.

*RESPONSE:*

*We have included the total hemispherical emittance.*

*REVISED TEXT in Supplementary Materials:*

**“At 0 °C, the hemispherically-weighted emissivity of the emitter in the atmospheric window (8 - 13 μm) is 0.632, while that outside the atmospheric window is only 0.086. This results in a total hemispherical emissivity of the emitter to be 0.247, since the blackbody fraction29 of the atmospheric window at 0 °C is 29.5%.”**

**(R1 query 15)** Line 387: Discusses the absorbed power from the atmosphere, while neglecting the effect of the mirror cone and shading device which help to inhibit some of the incoming radiation.

*RESPONSE:*

*We have included the effect of the mirror cone in an updated Supplementary Materials D.*

In summary I look forward to seeing this manuscript published once these comments have been addressed as i feel the paper will be of great interest to a broad audience.

~~~~~  
~~~~~ **Reviewer: 2** ~~~~~  
~~~~~

In the present manuscript, Chen et al describe theory and experiments to radiatively cool an object significantly below the ambient and in fact significantly below (water's) freezing (point) during both night and day cycling. By combining thermal design and photonic design, they reach an average temperature reduction of 37 C compared to the surroundings when cycled over 24 hours and peak cooling of 42 C during daytime illumination. The main advantage over their previous work, Ref[14], is the use of better thermal management to reduce conduction and convection. They also present a new photonic structure with more selective emissivity. Overall, the work is well presented and shows a novel result: a factor of 8x better daytime radiative cooling. I do have a few comments and clarification that should be addressed, as described below.

RESPONSE:

We thank the reviewer for his/her overall favorable assessment, and the constructive comments.

Please see detailed responses below.

(R2 query 1) "Ultrahigh-Performance" in the title seems somewhat arbitrary. "High-performance" is adequate.

RESPONSE:

We have deleted the "ultra" in the title.

(R2 query 2) The authors say that they estimate the parasitic heat transfer coefficient, h , but more details are needed for how this was done. Without knowing how this is actually calculated the reader merely has to take the authors at their words.

(R2 query 3) Similar to the previous comment, in section A of the Supplementary Materials, h is quoted in the range of 0.1-0.3 Wm⁻²K⁻¹, but in section D it is quoted as 0.2-0.4 Wm⁻²K⁻¹.

RESPONSE to QUERIES (2) and (3):

We thank the reviewer for carefully checking our thermal design, and catching the inconsistency in estimating the parasitic heat loss. We refined our estimate of h to be in the range of 0.2-0.3 Wm⁻²K⁻¹.

We have added the details in Supplementary Materials A on how we estimate h . We have also updated the theoretical calculations (the blue shaded band) in Fig. 4 b-c to reflect this change.

(R2 query 4) It is mentioned that the selective emitter is composed of 700 nm of Si (see figure 3). Is the silicon crystalline or amorphous. I believe it is amorphous based on the deposition, but it should be stated as the dielectric properties could be different.

RESPONSE:

We have clarified this in the main text.

REVISED TEXT:

"It consists of layers of silicon nitride (Si_3N_4), amorphous silicon (Si), and aluminum (Al), with thickness of 70 nm, 700 nm, and 150 nm, respectively"

(R2 query 5) Figure 4b shows the change in temperature with Dew Point. Was the dew point measured or controlled? How?

RESPONSE:

The dew point is obtained from an online weather station located at Stanford.

We have clarified this in the main text.

REVISED TEXT:

"In Fig. 4b (4c) we consider the temperature reduction as a function of dew point (ambient air temperature), while keeping the ambient air temperature (dew point) fixed. Note that the dew point is obtained from a weather station at Stanford²³."

(R2 query 6) There has been significant renewed interest in radiative cooling over the last few years, and the manuscript could benefit from mentioning some of the recent advances outside of building cooling. Examples include radiative cooling of solar cells [see: Zhu Optica 1, pp. 32-38 (2014) and Safi, Optics Express, pp. A1120-A1128 (2015)], reaching cryogenic temperatures in space [Youngquist, Optics Letters 41, pp. 1086-1089 (2016)], and radiative cooling of an air-cooled solar-thermal power cycle [Zeyghami, Energy Conversion and Management 106, pp. 10-20 (2015)].

RESPONSE:

We thank this reviewer for pointing out these new references to us. We have added them to the main text.

~~~~~  
~~~~~ **Reviewer: 3** ~~~~~  
~~~~~

This study theoretically explores the practical limits of radiative cooling and experimentally demonstrates a radiative cooling system with much-improved performance. The paper is very well written and well put together. The results are clear, and the methodology sound and well outlined.

*RESPONSE:*

*We thank the reviewer for the favorable evaluation of the technical quality of the work.*

My main concern, though, is the novelty and the impact of the work, in particular, whether it warrants publication in Nature Communications. When it comes to radiative cooling, the results from the previous work from the same authors (ref 17) are both intriguing and very relevant. In [17] the same authors showed something that was not at all expected or even intuitive (at least to a non-specialist): namely, they showed that daylight radiative cooling is possible in a system with a relatively rudimentary thermal insulation from the environment (and certainly in a system without any evacuation of the surrounding air).

With this result established, the fact that an object should radiately cool even more in vacuum (that is, when the heat conduction channel is further suppressed) is quite straightforward. Even though the emitter structure in [17] is somewhat different from the one in this work (for example, it seems to have higher emissivity above 15um), I think the authors would agree that the structure from [17] would also substantially cool down if also placed in a vacuum environment.

It implies that, by and large, the main contributor to added cooling performance (and the main difference from previous works) is the introduction of vacuum to suppress parasitic heat losses. To me this is mostly a technical improvement, meaning that the paper seems short of reaching the novelty and the impact threshold needed for publication in Nature Communications. Additional means or reducing the heat loss have either been used for radiative cooling previously (e.g. acrylic/polystyrene to reduce heat conduction in [17] similar to ceramic pegs here) or are industry standards (e.g. radiation shields/shades/mirrors).

*RESPONSE:*

*We appreciate the very kind words of the reviewer about our previous work as reported in Ref. [17]. When Ref. [17] was reviewed, several referees actually felt that paper was obvious or incremental. The point, of course, is that whether a paper is "straightforward" or not can be quite subjective.*

*While we certainly respect the view of the reviewer, we would like to take this opportunity to highlight the contributions of this paper. The main contributions are to theoretically determine the performance limit for radiative cooling in a typical populous environment, and to experimentally demonstrate a pathway towards achieving such a limit. As*

*a result of such combined theory and experimental efforts we were able to achieve a record performance in radiative cooling. These contributions are new - they have never been previously reported in the literature.*

*In terms of impact, as pointed out by Reviewer 2, "there has been significant renewed interest in radiative cooling in the last few years", in part because radiative cooling points to the exploration of a huge thermodynamic resource, i.e. the coldness of the universe, that has been largely under-utilized for renewable energy applications. In this context, our demonstration of record performance should generate substantial interests from both the technical community as well as the general public, and provide a significant stimulus to push the field forward. As such we do believe that the work should be published in Nature Communications.*

I would like to offer several comments and additions that may improve an otherwise very sound study. While these additions won't necessarily change the issues of novelty and impact, I hope the authors might find them useful.

*RESPONSE:*

*We thank this reviewer for these helpful comments. Please see our detailed responses below.*

**(R3 query 1).** Figure 2c very nicely shows the difference between an ideal ( $h=0$ ) and a non-ideal ( $h=8$ ) parasitic losses case. It would be helpful to also show some intermediate cases of  $0 < h < 8$ , to allow for visualization of the progression of improved cooling with the reduction in heat loss. This could maybe be an additional supplementary figure.

*RESPONSE:*

*We have added a supplementary figure (Fig. S5) and a corresponding section (Supplementary Materials G) to include more  $h$  values.*

**(R3 query 2).** A comment on the relationship between the vacuum pressure and the value of  $h$  (or, maybe an estimate for their system) could be useful.

*RESPONSE:*

*The total parasitic heat loss of our system comes from three parallel channels: the air conduction, the radiation from the backside of the selective emitter, and conduction through ceramic pegs. Vacuum only helps with the air conduction.*

*In Supplementary Materials A, we have added comment on the relationship between the vacuum level and the parasitic heat transfer coefficient through air conduction,  $h_{air}$ .*

**(R3 query 3).** The authors focus on the equilibrium temperature of the cooled device. However, an analysis of radiative cooling power (or extracted heat power) would be a nice addition to this work. If this is not immediately applicable to the geometry of their system, the authors

may consider alternative cases where the cooling emitter is in direct contact (at least on one side) with the object intended to be cooled down.

*RESPONSE:*

*The selective emitter in this work is designed to reach a record low temperature under the condition of low cooling loads. For applications with high cooling loads, a different selective emitter should be designed.*

*We have added a supplementary figure (Fig. S4) and a corresponding section (Supplementary Materials F) to clarify the different needs for achieving low temperature versus achieving high cooling power.*

**(R3 query 4).** (minor) On line 207, a system of "solar water collectors" is referenced. A direct reference (instead of a reference of a reference, such as [23]) would improve "bookkeeping", but this is a minor point.

*RESPONSE:*

*We have deleted ref. 23, and only kept ref. 22 (now 26 in the revised manuscript) which is a direct reference.*

Reviewer #1 (Remarks to the Author)

The authors have satisfactorily responded to each of the reviews comments/suggestions with numerous modifications made which significantly enhance the quality of the paper. I feel the paper should be accepted.

Reviewer #2 (Remarks to the Author)

The authors have addressed my concerns. In my opinion, the paper is ready for publication.

Reviewer #3 (Remarks to the Author)

I do not have additional comments for the authors. I still feel the conceptual novelty is limited, but I do recognize that the authors have addressed the technical questions appropriately.